# Equity in walking access to community home care facility resources for elderly with different mobility: A case study of Lianhu District, Xi'an

**Wenze Ning**, **Yi Yang**, **Mei Lu***, **Xiaokang Han**

College of Management, Xi'an University of Architecture and Technology, Xi'an, China

* lumei@xauat.edu.cn

## Abstract

As the aging of China's population continues to deepen, a number of elderly care facilities relying on community platforms to provide home care services have been established in urban communities, effectively alleviating the problem of difficult community elderly care, while a spatial mismatch between the facilities and the elderly population has also emerged. To solve this problem, this paper analyzes the equity in walking access to community home care facilities for elderly people with different mobility abilities in Lianhu District of Xi'an City, taking the resources of community home care facilities as the research object. Firstly, the coverage rate of the facilities was calculated based on the 15-minute walking range of the elderly with different mobility, and the accessibility of the facilities was measured using the Kernel Density-type two-step moving search method. Then, Gini coefficient, Lorenz curve and location entropy were used to analyze the spatial matching pattern of facilities and elderly population. The results show that there is a serious spatial mismatch between the resources of community home care facilities and the elderly population with mobility restriction. In addition, the available facility area per capita is low for more than 80% of the elderly with mobility restriction, and the road network density has a significant impact on the access of the elderly with mobility restriction to the community home care facility resources. These research results indicate that the spatial layout and configuration of community home care facilities are unfair to the elderly with poor mobility, and that these elderly care facility configurations do not favor the disadvantaged groups.

## 1. Introduction

With the development of society, the average life expectancy of the population increases, the low fertility rate and severe childlessness have led to the continuous deepening of aging in China. Xi'an is an important central city in western China. Data from the 7th National Census shows that Xi'an has 2,075,300 elderly people aged 60 and above, accounting for 16.02% of the city's total population. Geographic location is one of the important factors influencing elderly

**Data Availability Statement:** All relevant data are within the paper and its Supporting Information files.

**Funding:** This research was funded by the Key Research of Shaanxi Provincial Department of

Education (21JZ034), and the Ministry of Education Humanities and Social Sciences Planning Foundation (21YJA630092). The funder of the Ministry of Education Humanities and Social Sciences Planning Foundation (21YJA630092) had no role in study design, data collection and analysis, decision to publish, or preparation of the manuscript. The funder of the Key Research of Shaanxi Provincial Department of Education (21JZ034) supported writing the paper.

**Competing interests:** The authors have declared that no competing interests exist.

people's choice of elderly care, and the vast majority of elderly people in China choose to age in a familiar environment, implying that most urban elderly people choose to age at home in the community [1,2]. The 'Xi'an City Senior Care Plan 2018-2030' and the '15-minute community home care Senior Care Circle' program clearly point out that by the end of 2021, the city's senior care facilities will cover 100% of urban communities and achieve a fair spatial layout of senior care facilities. Therefore, the construction of facilities has become a key issue at present. The geographical space of facilities itself is different, and the needs of elderly people vary from person to person. Therefore, the goal of community home care facilities planning is to meet the access of different groups of elderly people to equal facility resources, rather than to meet the equity of geographic space. Accurate analysis of the equity of different mobility elderly people's access to the resources of community home care facilities in their living environment can provide reference for the layout of urban community elderly care home facilities.

Facility layout equity refers to the equal access to facility resources based on quantitative and spatial matching [3]. Regarding the research on facility layout equity, scholars mainly focus on the rational degree of facility layout, aiming to reduce the unequal supply of facility resources caused by class differences and make the facility configuration and layout more inclined to the disadvantaged groups [4]. Scholars initially focused on health care facilities and educational facilities, in recent years there has been more research on senior living facilities and the more novel parkland. Research questions have gradually advanced from accessibility to equity. For the study of senior care facilities, scholars took the number of beds per capita in senior care facilities as the research index, and used the 2SFCA, Lorenz curve, and Locational entropy to conduct equity analysis, and the study emphasized the differences in the resources of senior care facilities per capita in each region [5]. For the study of educational facilities, scholars use the radius of school services and the radius of 15-minute living circle as accessibility evaluation indexes, and the results show that the road network is well developed in the urban core, the number of schools is large, the accessibility of neighborhoods near the core is high, and the accessibility of peripheral neighborhoods far from the core needs to be improved [6]. For the study of medical facilities, scholars took the distance from residential points to medical facilities as the accessibility evaluation index and used the 2SFCA for accessibility analysis, compared with the traditional method of the ratio of medical personnel to the population of residential points, the 2SFCA can show the spatial variation of the accessibility of medical facilities in the study area, and the results show that the accessibility of areas along transportation routes is higher [7]. For the study of park green space, scholars take the per capita park green space area and coverage rate as evaluation indexes, and consider the spatial distribution of user demand, so as to reflect the spatial match between park green space and users before [8]. By combing scholars' literature, it can be concluded that the methods for analyzing the accessibility and equity of public service facilities mainly include the 2SFCA method, Lorenz curve method, locational entropy method, and the factors affecting the equity of facilities include the spatial distribution characteristics of different demand groups, the accessibility distribution characteristics of facilities and road network density.

Existing studies generally evaluate the equity of facility layout by analyzing facility layout characteristics, facility accessibility, and the degree of matching population and facility resources. Tang *et al.* used the nearest neighbor index method to analyze the types of spatial distribution of elderly facilities in the Changsha-Zhuzhou-Xiangtan urban cluster and the kernel density estimation method to analyze the clustering characteristics of elderly facilities, concluding that elderly population distribution, accessibility and medical facilities are the factors affecting the spatial differentiation of elderly facilities The main factors [9]. Based on a GIS platform, Zhao *et al.* studied the spatial distribution of elderly and home-based elderly care service facilities in Shahekou District, Dalian City, and used 2SFCA to calculate the supply and

demand ratio of the elderly population to facility resources, and derived accessibility differences in 89 communities, which provided support for the development of community-based elderly care facilities [10]. R G and G C summarized the studies on the planning and layout of elderly facilities, which mainly focused on the field of elderly geography, exploring the equilibrium of the spatial distribution of elderly resources at the national and regional levels, with few studies at the level of urban and rural planning strategies [11]. Ryvicker *et al.* studied the factors that facilitate and hinder elderly people's access to elderly resources, and the results showed that disabled elderly people are very sensitive to geographical and spatial barriers, and that spatial distance and transportation modes have an important impact on elderly people's access to elderly resources, thus indicating the importance of the layout of elderly facilities [12]. Tao *et al.* used a modified 2SFCA to assess the spatial accessibility of senior care facilities in Beijing, taking into account the distance attenuation factor, and concluded that the spatial imbalance of senior care facilities in Beijing is not balanced. He concluded that there is a severe shortage of land for the construction of senior care facilities in the central areas of the city, and that senior care facilities should be located in the suburbs to serve the elderly in the central city [13]. Wang *et al.* argued that previous studies were only at the street level and administrative district level, which would lead to inaccurate results on the accessibility of elderly people to community elderly facilities. She used 2SFCA method to measure the accessibility of community elderly facilities based on 3,494 community POI data in Guangzhou, and the results showed that the distribution trend of accessibility of community elderly facilities in Guangzhou was opposite to the distribution trend of elderly population density. The accessibility of facilities is poor in urban centers where the elderly population is dense, and high in peripheral areas where the elderly population is sparse. She suggests that the accessibility of community elderly facilities should be improved by making full use of existing community elderly facilities and reorganizing and renovating them [14].

Through combing the existing literature, it is found that most of the current studies are based on the perspective of all elderly people, and there are fewer studies based on the perspective of elderly people with different physiological differences. Elderly people with poor health conditions should be high frequency users of community home care facilities, and attention should be paid to the equity of access to community home care facilities resources for such elderly people. Secondly, most studies use Gravity-type 2SFCA and Gaussian-type 2SFCA to measure the accessibility of medical and sports facilities and park green space, but according to the research results, the elderly's willingness to travel to community home care facilities decreases slowly at first and then sharply with the increase of travel distance, and this decreasing trend is not consistent with the function curves of Gravity-type 2SFCA and Gaussian-type 2SFCA, but with the kernel density function.

In view of this, this paper attempts to change the research approach based on the overall elderly perspective to study the equity of access to community home care facilities resources for elderly people with different mobility abilities. The fairness evaluation system proposed in this paper is not the traditional evaluation of the fairness of each type of elderly people's access to all facility resources individually, but the evaluation of facility fairness in the case of simultaneous access to facility resources by elderly people with different mobility. Taking the Lianhu District of Xi'an City as an example, this paper analyzes the spatial distribution characteristics of the elderly population and community home care facilities in Lianhu District based on a GIS platform, and then classifies the elderly into elderly with normal mobility, elderly with mild mobility restriction, and elderly with severe mobility restriction by investigating the walking speed of the elderly. The service capacity of the community home care facilities was analyzed according to the 15-minute walking distance of the elderly with different mobility; then the accessibility of the community home care facilities was measured using the Kernel

Density-type 2SFCA based on the actual road network; then the Gini coefficient method, the Lorenz curve method, and the Locational entropy method were used to analyze the social equity performance. By using this series of methods to analyze the fairness of the layout of community home care facilities from the perspective of elderly people with different mobility, we grasp the differences in access to community home care facilities resources for elderly people with different mobility. Finally, discussions and conclusions of this study are presented based on the analysis results. In order to promote equitable access to community home care facilities resources for elderly with different mobility and provide a reference for community home care facilities construction.

## 2. Materials and methods

### 2.1. Study area

The study area of this paper is the Lianhu District of Xi'an and the nine streets under its jurisdiction, see Figs 1 and 2. Lianhu District is one of the old urban areas of Xi'an, with an administrative area of 43 km$^2$ and a resident population of 1,019,100 at the end of 2020. In this article, the map images and road network vector data for the Lianhu District of Xi'an are from the OpenStreetMap (OSM) open source wiki map (https://www.openstreetmap.org/), and are not copyrighted.

### 2.2. Data sources

The administrative boundary data of Lianhu District, Xi'an City used in this study were obtained from the National Public Geospatial Information Service Platform, while the vector data of Xi'an City roads were obtained from the OSM website, and the single-lane road network of Lianhu District was obtained by performing buffer setting, double-lane to single-lane and clipping operations in ArcGIS 10.2 software. By crawling the POI data of second-hand

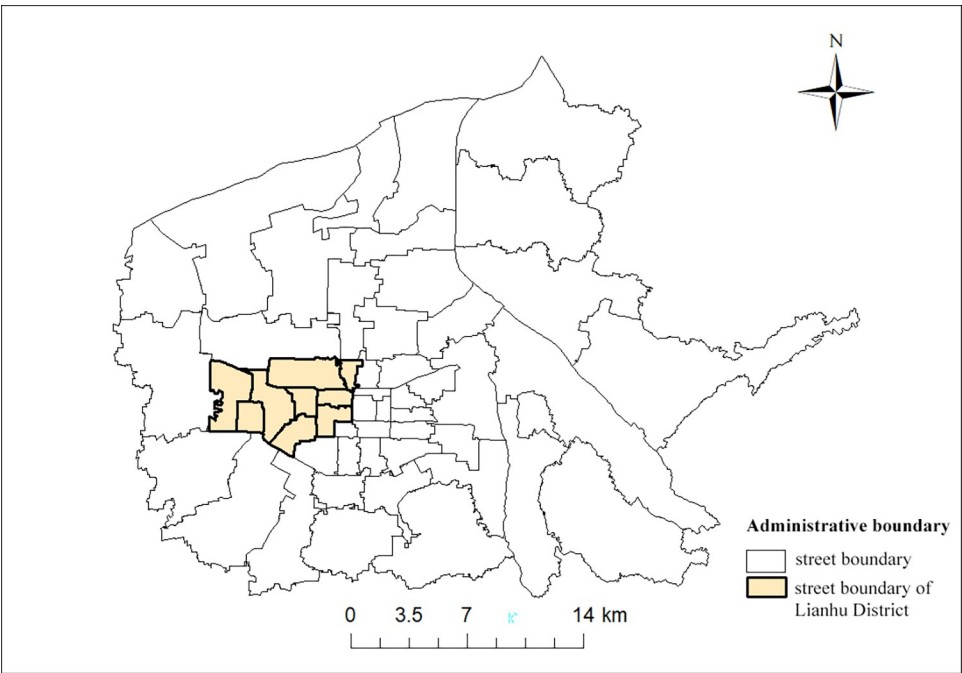

**Fig 1. Six Districts of Xi'an City.** (https://www.openstreetmap.org/relation/3226093).

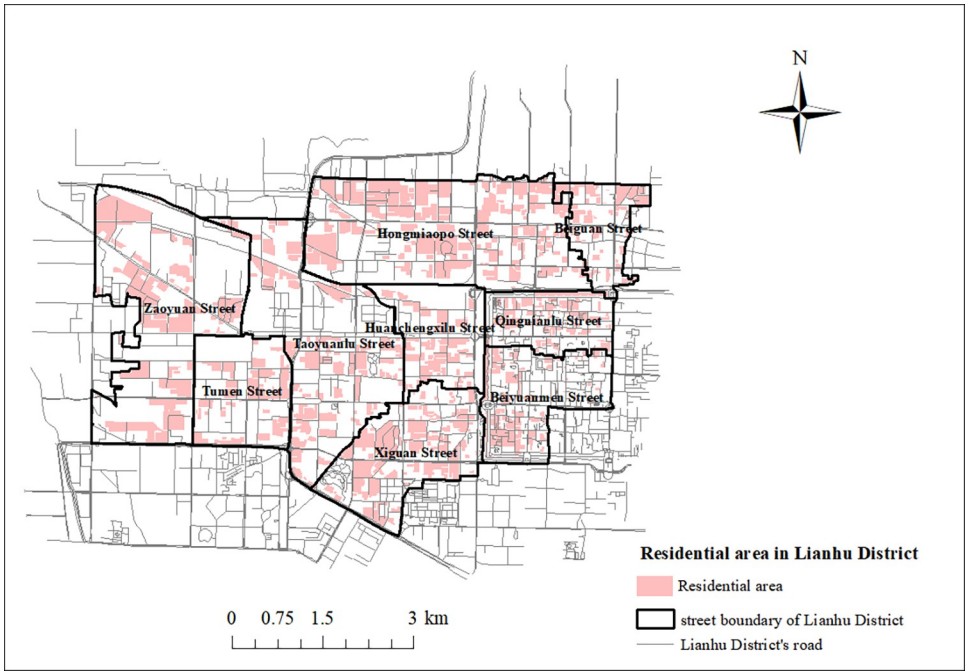

**Fig 2. Street boundary map of Lianhu District.** (https://www.openstreetmap.org/relation/3226093).

housing plots in the rental housing network information of Xi'an City Lianhu District, we obtained the information of plot names and number of households, calculated the number of elderly people living in each plot using the average household population data from the latest street-level statistics of the 7th census of Xi'an City Lianhu District in 2020, and corrected the plot population with the 7th census data for the street-level data. We obtained the names, locations and facility areas of community elderly at home facilities in Lianhu District by searching the official website of the Civil Affairs Bureau of Lianhu District, Xi'an, and used the Baidu coordinate picking system to obtain the Baidu coordinates of all facilities, and then used QGIS3.16 software to convert the BD09 coordinates of all communities and community elderly at home facilities obtained into WGS1984 geographic coordinates, and after projection, we established the projections of the communities, the GIS spatial database of elderly population and community home-based elderly facilities in Lianhu District after projection.

**2.2.1. Facility data.** The community home care facilities studied in this paper include community home care service stations, day care centers, elderly restaurants and elderly activity centers, as shown in Table 1. According to the announcement of Xi'an Lianhu District Civil Affairs Bureau, by the end of 2020, there are 117 community home care service stations, 13 day care centers, 18 elderly restaurants and 18 Senior centers in the normal operation community of Lianhu District. The spatial distribution is shown in Fig 3. The information on senior care facilities used in this article was obtained from the information on community home care facilities made public by the Lianhu District Civil Affairs Bureau in 2021 (**http://www.lianhu. gov.cn/**). The GIS platform is used to analyze the overall spatial distribution characteristics of these four types of facilities. The results show that the nearest neighbor index is less than 1, and the spatial distribution presents a state of aggregation.

**2.2.2. Community data.** The data on the residential locations in Lianhu District used in this article were obtained from Shell (https://xa.ke.com/), a website that allows you to search the geographical location of all neighborhoods in Lianhu District for free. In this study, we

**Table 1. An overview of Xi'an community home care facilities.**

| Name of community home care facilities | Number of facilities | Service items |
|---|---|---|
| Community home care service stations | 117 | Community home care service station is responsible for organizing service personnel to provide leisure and entertainment, catering, psychological comfort and other services for the elderly in the community. |
| Day care center | 13 | Day care centers focus on semi-disabled and disabled elderly people, providing daily care, catering, cultural entertainment, psychological comfort and other services. |
| Elderly restaurants | 18 | Restaurant for the elderly provides catering services for the elderly in the community according to their eating habits. The elderly may choose to eat in the store or serve at the door. |
| Senior center | 18 | The Senior center sets up different service spaces according to the different needs of the elderly. The service space can be used for the elderly to carry out chess, vocal music, dance and other activities. |

obtained information about communities in Lianhu District form second-hand rental web-sites, searched 825 communities, and recorded the name, number of households, and address of each community. Based on the collected addresses of these 825 communities to search their geographic coordinates, a GIS database is created and these communities are marked on the GIS. The administrative map layer of Lianhu District in the GIS overlaps with the community layer, and the intersection tab operation in ArcGIS10.2 adds the street information of the communities in the community attribute table and exports the table. In the table, the average household population of each street level in Lianhu District from the 7th National Census of China published in 2020 was multiplied by the number of households in the community of the street to which it belongs, and the population of each community was obtained. Considering the existence of vacant houses and in order to make the population number calculation more

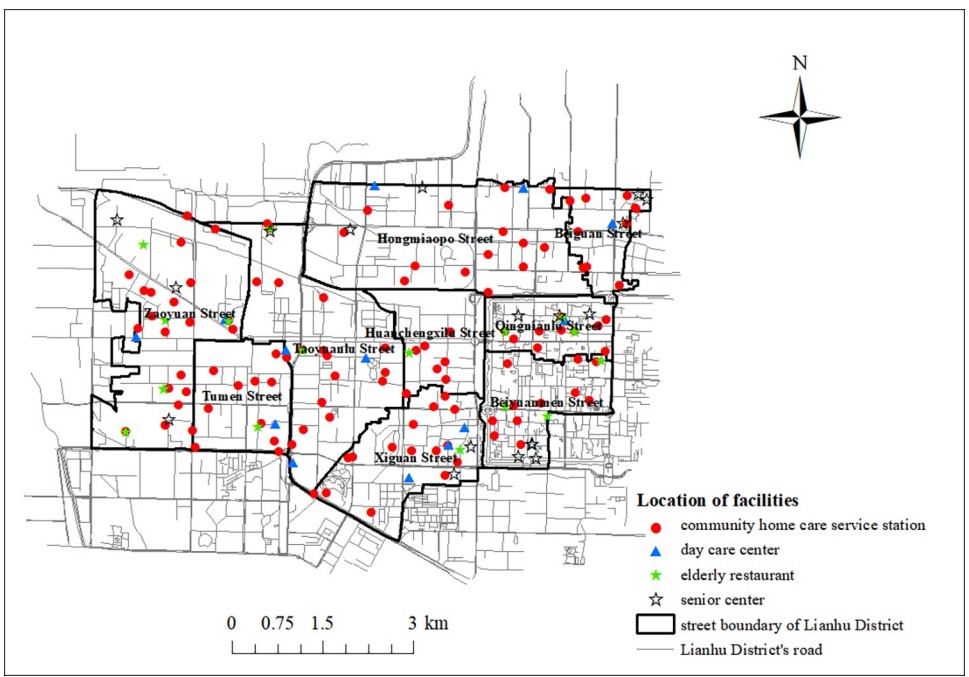

**Fig 3. Spatial distribution of community home care facilities.** (https://www.openstreetmap.org/relation/3226093).

accurate, we calculated the sum of the population numbers of the subdivisions in each street in Lianhu District, and compared the population numbers of each street level in Lianhu District from the data of the 7th National Census with the population numbers of each street calculated in this study to obtain a ratio. This ratio was multiplied by the population size of each community to obtain the population size of each community after correction for street-level census data. Finally, the aging rate of each street level in Lianhu District from the seventh national census data was multiplied by the population size of each community in the respective street to obtain the number of elderly people in each community, as shown in **Fig 4**.

**2.2.3. Elderly population data.** The population data used in this paper are from the Seventh National Census (2020) data, which is publicly available from the government (http://www.stats.gov.cn/tjsj/pcsj/), and is the most current data available. With a population of 182,300 people aged 60 and above within the 7th National Census of 2020 in Lianhu District, its elderly population tops the six districts of the main city of Xi'an. In terms of street-level data, the streets with a larger elderly population are Hongmiaopo Street, Taoyuanlu Street and Zaoyuan Street, all of which have an elderly population of over 20,000. The overall aging rate in Lianhu District is 17.89%, higher than the Xi'an city average of 16.02%. All nine streets under the jurisdiction of Lianhu District have an aging rate of more than 14%. Among them, the streets with a higher degree of aging are Beiyuanmen Street and Qingnianlu Street, whose aging rates are 23.44% and 24.30% respectively, the two streets with the highest degree of aging among the 57 streets in the six districts of the main city of Xi'an. The area of Lianhu District is 5.09% of the area of the six districts of the main city of Xi'an, but its elderly population aged 60 and above accounts for 22.10%, which shows its highly dense elderly population. The elderly population density of nine streets in Lianhu District exceeds 3,000 people per square kilometer, among which, the elderly population density of Beiguan Street even reaches 7,946 people per square kilometer, which is the largest among the 57 streets in the six administrative districts of Xi'an city center, as shown in Fig 5.

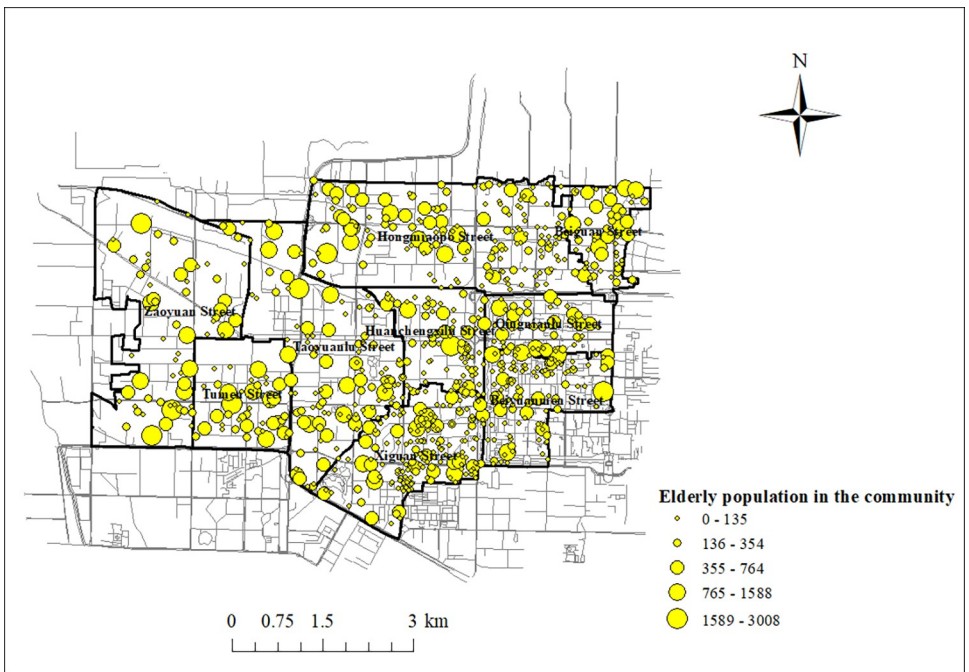

**Fig 4. Spatial distribution of elderly population in the community.** (https://www.openstreetmap.org/relation/3226093).

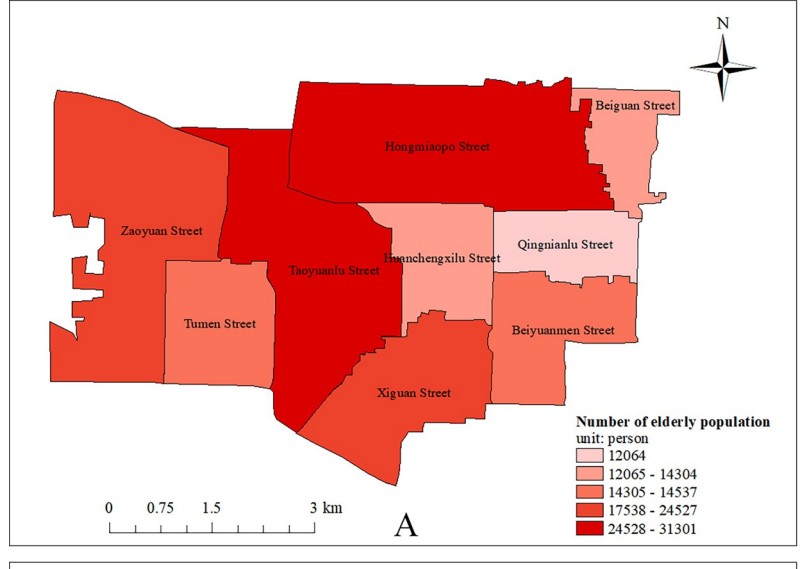

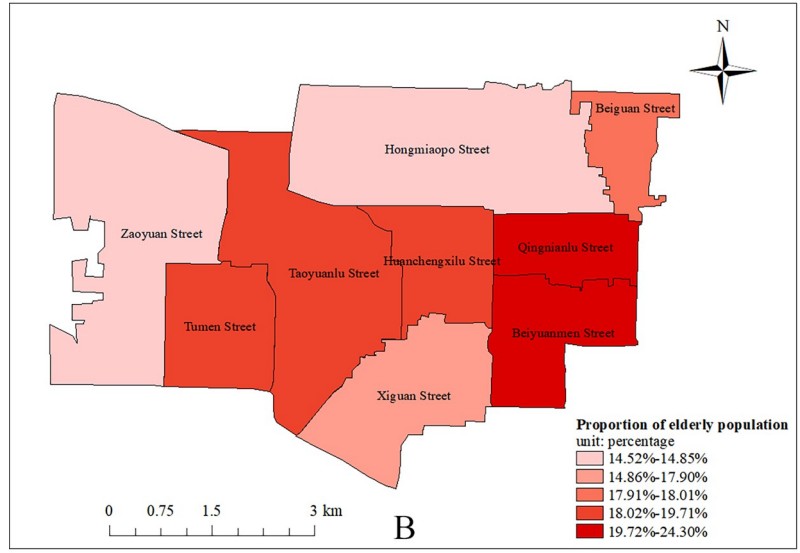

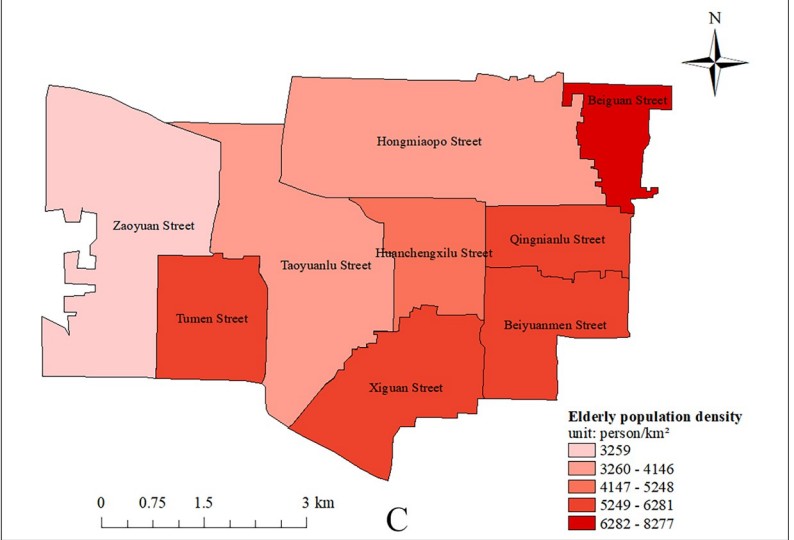

**Fig 5. Overview of the elderly population in Lianhu District.** (A) Fig of number of elderly population; (B) Fig of aging rate; (C) Fig of the elderly population density. (https://www.openstreetmap.org/relation/3226093).

## 2.3. Methods

**2.3.1. Kernel Density-type two-step floating catchment area method.** Two-Step Floating Catchment Area Method (2SFCA) was first proposed by Radke *et al* [15]. 2SFCA considers both supply and demand, and analyzes the supply and demand in the study area, so it is widely used in the accessibility evaluation of public service facilities [16–19]. The traditional 2SFCA thinks that the accessibility value of facilities within the search threshold range is the same, but in reality, the residents' travel intention will decrease with the increase of travel distance. To solve this problem, scholars increase the distance attenuation function on the traditional 2SFCA model, and expand the Gravity-type, Gaussian-type and Kernel Density-type 2SFCA. The attenuation curves of accessibility under different types of 2SFCA are different [20], as shown in Fig 6. This paper uses the Kernel Density- type 2SFCA to study the accessibility of community home care facilities. The kernel density distance attenuation function is more in line with the travel willingness of the elderly. The greater the travel distance is, the faster the attenuation is, and vice versa [21]. Meanwhile, the mobility of elderly people is one of the important factors affecting the accessibility of facilities in this paper, and the degree of distance attenuation varies among the facilities that elderly people can reach within a 15-minute walk depending on their mobility, as shown in Fig 7.

The first step, for each facility point *j*, the space distance threshold is given to form a space scope; the number of potential users in the facility point *j* service radius can be obtained by weighting the population of each demand point *k* in the space domain using the kernel density equation and adding the weighted population; then the area of facility point *j* is divided by the number of potential users to obtain the supply-demand ratio $R_j$, the supply-demand ratio can be expressed as follows:

$$R_j = \frac{S_j}{\sum_{k \in \{d_{kj} \leq d_0\}} G(d_{kj}, d_0) P_k} \tag{1}$$

In the formula, $P_k$ is the population of each demand point *k* in the space scope of facility point($d_{kj} \leq d_0$); $d_{kj}$ is the space distance from the centre of demand point *k* to facility point *j*; $d_0$ is the travel limit distance; $S_j$ is the construction area of facility point *j*; $G(d_{kj}, d_0)$ is a kernel

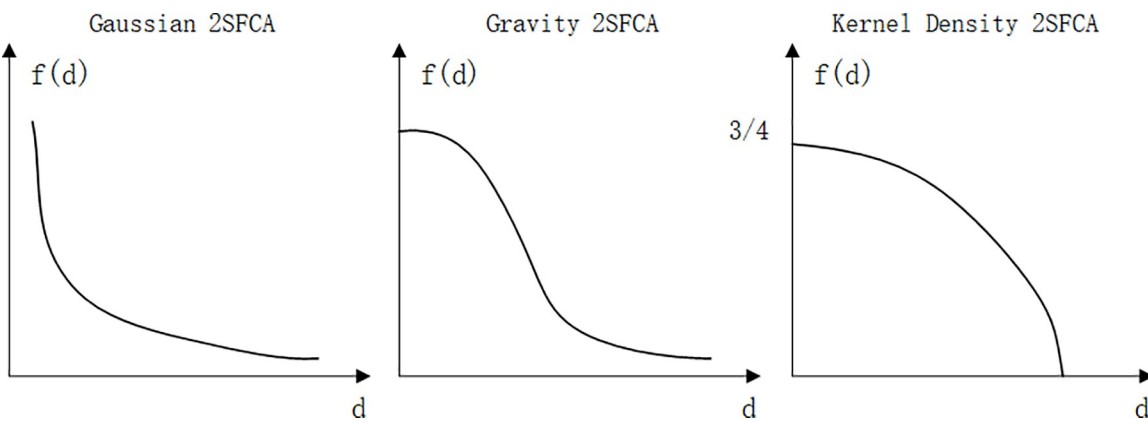

**Fig 6. Diagram of distance decay function.** (Self-drawn by the author).

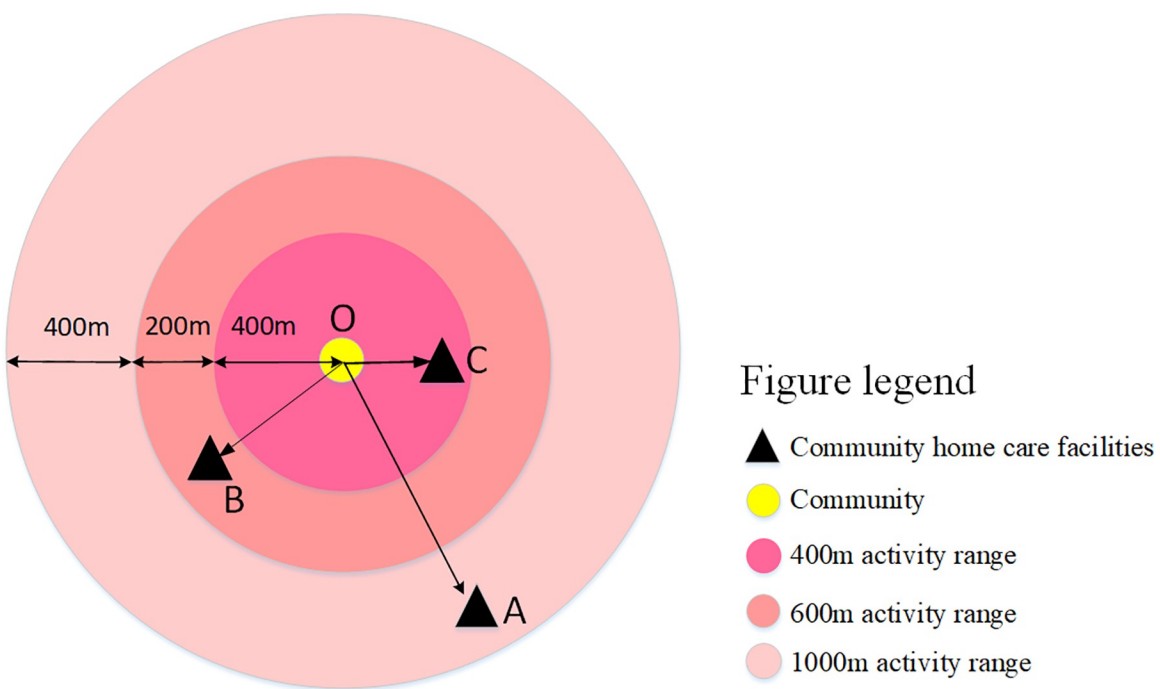

**Fig 7. Diagram of the ability of elderly people with different mobility to access community home care facilities.** (Self-drawn by the author).

density equation considering distance decay, it can be expressed as follows:

$$G\left(d_{kj}, d_0\right) = \frac{3}{4}\left[1 - \left(\frac{d_{kj}}{d_0}\right)^2\right], \left(d_{kj} \le d_0\right) \tag{2}$$

In the second step, for each demand point $k$, the threshold of spatial distance is given to form another spatial action domain $d_0$. For the supply and demand ratio of each facility point $j$ falling within the spatial action domain, the Gaussian equation is used to assign weights, and then these weighted supply and demand ratios are added to obtain the accessibility $A_i$ of demand point $i$, the accessibility value can be expressed as follows:

$$A_i = \sum_{j \in \{d_{kj} \le d_0\}} G(d_{kj}, d_0)R_j \tag{3}$$

In the formula. $A_i$ can be interpreted as the area per capita with access to community home care facilities in persons/m$^2$, and other indicators are described as in Eqs (1) and (2).

**2.3.2. Gini coefficient method and Lorenz curve method.** Gini coefficient and Lorenz curve are commonly used in economics to evaluate regional or national income distribution differences [22]. Due to the similarity of social equity connotation between income distribution and public resources distribution, scholars apply Gini coefficient and Lorenz curve to the performance evaluation of public transport, medical facilities, sports facilities and pension facilities [23–25]. This paper uses Gini coefficient to analyze the matching degree of community home care facilities resources and the elderly population. Lorenz curve was used to analyze the proportion of elderly people enjoying community home care facilities resources under different population proportions. The facility accessibility under 2SFCA represents the facility resources per capita. The facility accessibility values of each community are sorted, and the

elderly population is accumulated to draw the Lorenz curve. The Gini coefficient can be expressed as follows:

$$G = 1 - \sum_{k=1}^{n}(P_k - P_{k-1})(R_k + R_{k-1}) \tag{4}$$

In the formula: $G$ is the Gini coefficient of social equity performance of community home care facilities; $P_k$ is the cumulative ratio of the elderly population; $R_k$ is the cumulative accessibility ratio of community home care facilities; $k$ is community number. The $G$ value is between 0 and 1, the greater the value, indicating that the more uneven the distribution of community home care facilities resources, the worse the social equity performance, and vice versa.

**2.3.3. Location entropy.**  The Gini coefficient and Lorenz curve analyze the resource allocation of community home care facilities, but these two methods lack the specific spatial matching degree between facilities and the distribution of the elderly population. Therefore, the location entropy method is used to analyze the spatial pattern of social justice performance. Location entropy is one of the economic methods, which is often used to analyze the degree of specialization of leading industries in the region and help to evaluate the spatial concentration of a certain factor. In recent years, it is commonly used to evaluate the spatial pattern of public service facilities [26–28]. According to the principle of this method, this paper uses the location entropy to analyze the concentration degree of community home care facilities in various streets of Lianhu District. The location entropy of each street is the ratio of the area of the elderly per capita using community home care facilities in the street to the area of the elderly per capita using community home care facilities in the whole study area. The higher the value, the more concentrated the resources of home care facilities in the street community. The location entropy can be expressed as follows:

$$LQ_j = \frac{(T_j/P_j)}{(T/P)} \tag{5}$$

In the formula: $LQ_j$ is the locational entropy of each street; $T_j$ is the building area of community elderly home facilities in street $j$ and $P_j$ is the number of elderly people in street $j$; $T$ is the building area of community elderly home facilities in the whole study area and $P$ is the number of elderly people in the whole study area.

## 3. Results

### 3.1. Analysis of community home care service ability

Dai et al. introduced the concept of 'elderly with mobility restriction' to evaluate the difference in the frequency of using elderly facilities between the healthy elderly and the elderly with mobility restriction. In this paper, we further classified 'elderly with mobility restriction' into 'elderly with mild mobility restriction' and 'elderly with severe mobility restriction', and used unstructured observation method to investigate the gait speed of the elderly. We selected 36 locations in nine Streets of Lianhu District, including community home care service stations, day care centers, elderly restaurants, senior centers, squares, and gardens, the purpose was to ensure that the research targets were not only the elderly who went to the community home care facilities to receive services, but also included other elderly people, so that the sample was representative. At each location, a section of sidewalk was selected as an observation interval, and the length of each observation interval was measured and the time taken by the elderly to pass through the interval was recorded to calculate the walking speed of the elderly. The target population was elderly people aged 60 and above without mental illness and with clear

**Table 2. Questionnaire of the elderly's mobility in Lianhu District.**

| Administrative Street | Elderly with normal mobility | Elderly with mild mobility restriction | Elderly with severe mobility restriction |
|---|---|---|---|
| Qingnianlu Street | 42 | 7 | 3 |
| Beiyuanmen Street | 48 | 10 | 4 |
| Beiguan Street | 43 | 8 | 2 |
| Hongmiaopo Street | 50 | 9 | 3 |
| Huanchengxilu Street | 42 | 7 | 4 |
| Xiguan Street | 39 | 9 | 1 |
| Tumen Street | 43 | 4 | 2 |
| Taoyuanlu Street | 38 | 5 | 3 |
| Zaoyuan Street | 45 | 6 | 1 |
| total | 390 | 65 | 23 |
| percentage | 81.6% | 13.6% | 4.8% |

consciousness, as well as their family members. A total of 478 elderly people were surveyed, and the results showed that there were 390 elderly people with normal mobility, 65 elderly people with mild mobility restriction, and 23 elderly people with severe mobility restriction, as shown in Table 2. According to the findings of this paper and combined with the results of Chinese Longitudinal Healthy Longevity Survey (CLHLS), the proportion of elderly people with different mobility in Lianhu District was determined, 80% of elderly people in Lianhu District were with normal mobility, 16% were with mild mobility restriction, and 4% were with severe mobility restriction [29,30]. According to the measurement results, the step speed of the elderly with normal mobility, the elderly with mild mobility limitation, and the elderly with severe mobility limitation were set to 1.1m/s, 0.67m/s, and 0.44m/s.

In 2021, the Xi'an government released the 'Xi'an Action Plan for Promoting the High-quality Development of Pension Service', which is committed to creating a '15-minute pension circle' in the city, so that the elderly can obtain diversified pension services after 15 minutes of walking. This paper selects the 15-minute walking distance of the elderly with normal mobility, the elderly with mild mobility restriction and the elderly with severe mobility restriction as the radius of facility service. According to the walking speed of the three types of elderly, the 15-minute walking distance is determined to be 1000m, 600m and 400m, respectively. ArcGIS10.2 software was used to set up the buffer zone of community home care facilities for coverage analysis. When the normal elderly walked 15 minutes, the coverage rate of community home care facilities in Lianhu District was 100% and the overlap rate was 91.04%. When the elderly with mild mobility restriction walks for 15 minutes, the coverage rate of community home care facilities in Lianhu District is 97.69%, and the overlap rate is 77.21%. When the elderly with severe mobility restriction walks for 15 minutes, the coverage rate of community home care facilities in Lianhu District is 80.72%, and the overlap degree is 59.22%, as shown in Fig 8. Without taking into account the actual road network, the overall coverage rate of community home care facilities in the Lianhu District from the perspective of the elderly with normal mobility and the elderly with mild mobility restriction reached the value of 90% set by the Shaanxi provincial government, except for the elderly with severe mobility restriction, which did not reach this standard. However, the overlap of the service scope of these three categories of elderly people is greater than 50%, which is too large and causes a waste of facility resources.

As shown in Table 3, from the street-level coverage analysis, the coverage rate of community home care facilities in nine streets in Lianhu District from the perspective of the elderly with normal mobility is 100%, the coverage rate of facilities in all streets from the perspective of the elderly with mild mobility restriction is over 94%, and the coverage rate of facilities in all

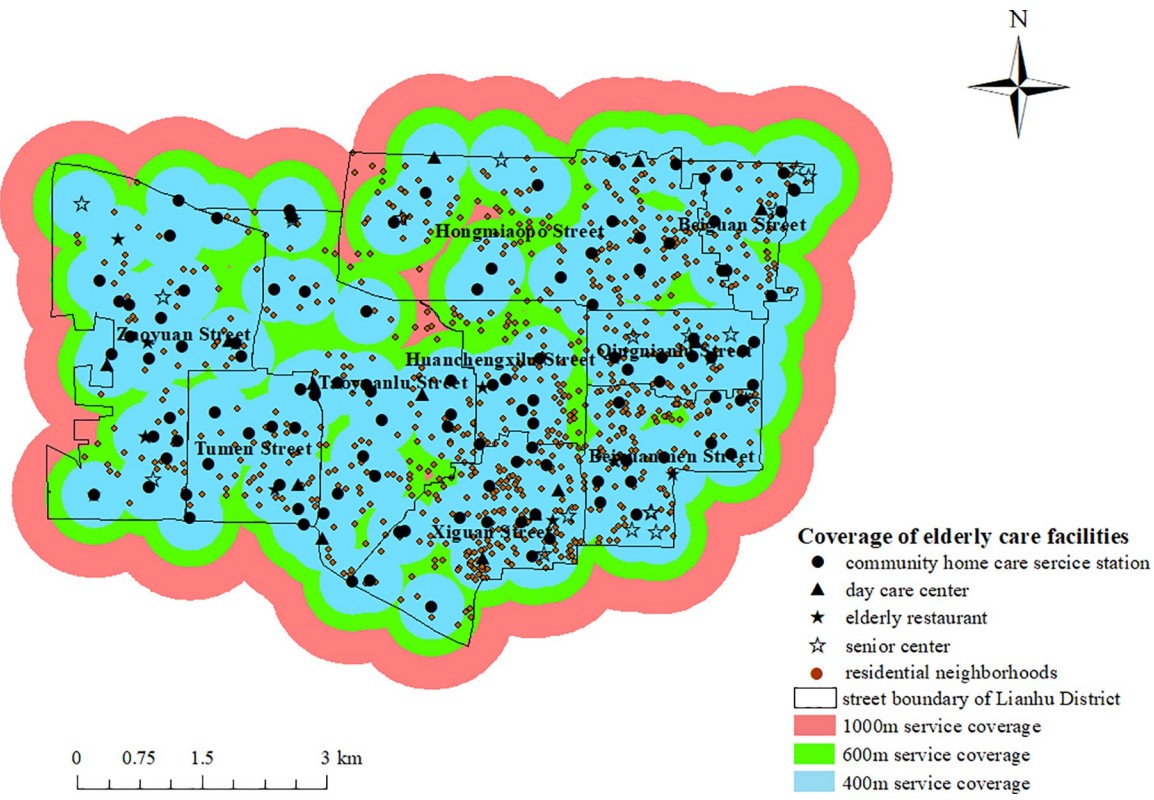

**Fig 8. The coverage rate of community home care facilities from the perspective of the elderly with different mobility.** (Self-drawn by the author).

streets from the perspective of the elderly with severe mobility restriction is the highest at 99.27% and the lowest at 67.99%, with obvious differences in coverage rates. Among them, Hongmiaopo Street, Taoyuanlu Street, Huanchengxilu Street and Zaoyuan Street have the lowest coverage rate of community home care facilities. The three reasons for the large area of the streets, the small number of facility points and the dense facility points lead to the low coverage rate of facility service scope in these streets. The most obvious street with dense facility points is Huanchengxilu Street. Huanchengxilu Street has a small area, but the community home care facilities in its streets are concentrated in the central and southern parts, resulting in low service coverage in other areas.

### 3.2. Accessibility analysis of community home care facilities

Based on the actual road network, ArcGIS10.2 software was used to establish the OD cost matrix and calculate the accessibility of the three types of elderly people walking 15 minutes to the community home care facilities in 825 communities, and the accessibility values were spatially visualized and analyzed by using Kriging interpolation method, as shown in Fig 9.

The results of the analysis showed that the overall accessibility of the elderly with normal mobility was better than the other two categories of elderly, but the accessibility of each community varied greatly, with the lowest accessibility value being 0 and the highest being 7.90, and the mean accessibility value being 0.20. 592 of the 825 communities in the study area, accounting for 71.67% of the total number of communities, had accessibility values lower than the mean value. This indicates that the influence of the scale of facilities, the density of facility distribution, and the degree of aggregation of the elderly population makes the accessibility of

**Table 3. Coverage rate of community home care facilities in Lianhu District.**

| Administrative Street | 1000 meters | 600 meters | 400 meters |
|---|---|---|---|
| Qingnianlu Street | 100.00% | 100.00% | 99.27% |
| Beiyuanmen Street | 100.00% | 100.00% | 95.27% |
| Beiguan Street | 100.00% | 100.00% | 95.92% |
| Hongmiaopo Street | 100.00% | 94.38% | 67.99% |
| Huanchengxilu Street | 100.00% | 99.41% | 68.21% |
| Xiguan Street | 100.00% | 100.00% | 92.04% |
| Tumen Street | 100.00% | 98.65% | 90.40% |
| Taoyuanlu Street | 100.00% | 96.78% | 76.10% |
| Zaoyuan Street | 100.00% | 97.71% | 78.28% |
| Lianhu District | 100.00% | 97.69% | 80.72% |

settlements in some areas vary significantly from the perspective of the elderly with normal mobility, and the overall level of accessibility is low, especially in the middle of Zaoyuan Street, the north of Hongmiaopo Street, and the south of Beiyuanmen Street. The accessibility of the elderly with mild mobility restriction and the elderly with severe mobility restriction were similar overall, with 86% of the total number of communities having a accessibility of 0-0.2, but the accessibility of the elderly with mild mobility restriction varied more, with the highest value of 5.60, while the highest value of 1.47 for the elderly with severe mobility restriction. The accessibility of the elderly with mild mobility restrictions was 0 in 92 settlements in the study area, accounting for 11.13% of the total number of settlements, while the accessibility of the elderly with severe mobility restriction was 0 in 256 communities, accounting for 30.99% of the total number of communities. These two types of elderly with mobility restriction are also affected by the scale of facilities, the density of facility distribution, and the degree of concentration of the elderly population, and the accessibility of community home care facilities varies significantly, and the overall level is low, and there are many blind service areas. At the same time, from Fig 9B and 9C, we can see that the road network density in the central part of Zaoyuan Street is low, and the accessibility value of the central part of Zaoyuan Street is as high as 5.6 from the perspective of the elderly with mild mobility restriction, and the highest accessibility value is 0.4 from the perspective of the elderly with severe mobility restriction, and the difference between the extreme travel distance of these two types of elderly with mobility restriction is 200 meters, but the accessibility value is very different, which indicates that the road network density is also one of the important factors affecting the accessibility of the facilities. This indicates that the density of the road network is also an important factor affecting the accessibility of facilities.

### 3.3. Social equity performance analysis of community home care facilities

The Gini coefficients of community elderly care facilities in each street office in Lianhu District are higher than those of the other two categories from the perspective of the elderly with normal mobility, and the Gini coefficients are less than 0.6, as shown in Table 4. As shown in Table 5, according to the Gini coefficient ranking stipulated by the United Nations Development Program, the Gini coefficient of the layout of community home care facilities in each street office in Lianhu District from the perspective of the elderly with normal mobility is higher than the other two categories of elderly, and the Gini coefficient is less than 0.6. The Gini coefficients of Qingnianlu Street and Beiguan Street are 0.27 and 0.29, which are in the 'comparative average' class. The Gini coefficients of the layout of community home care facilities in each street office from the perspective of the elderly with mild mobility restriction and

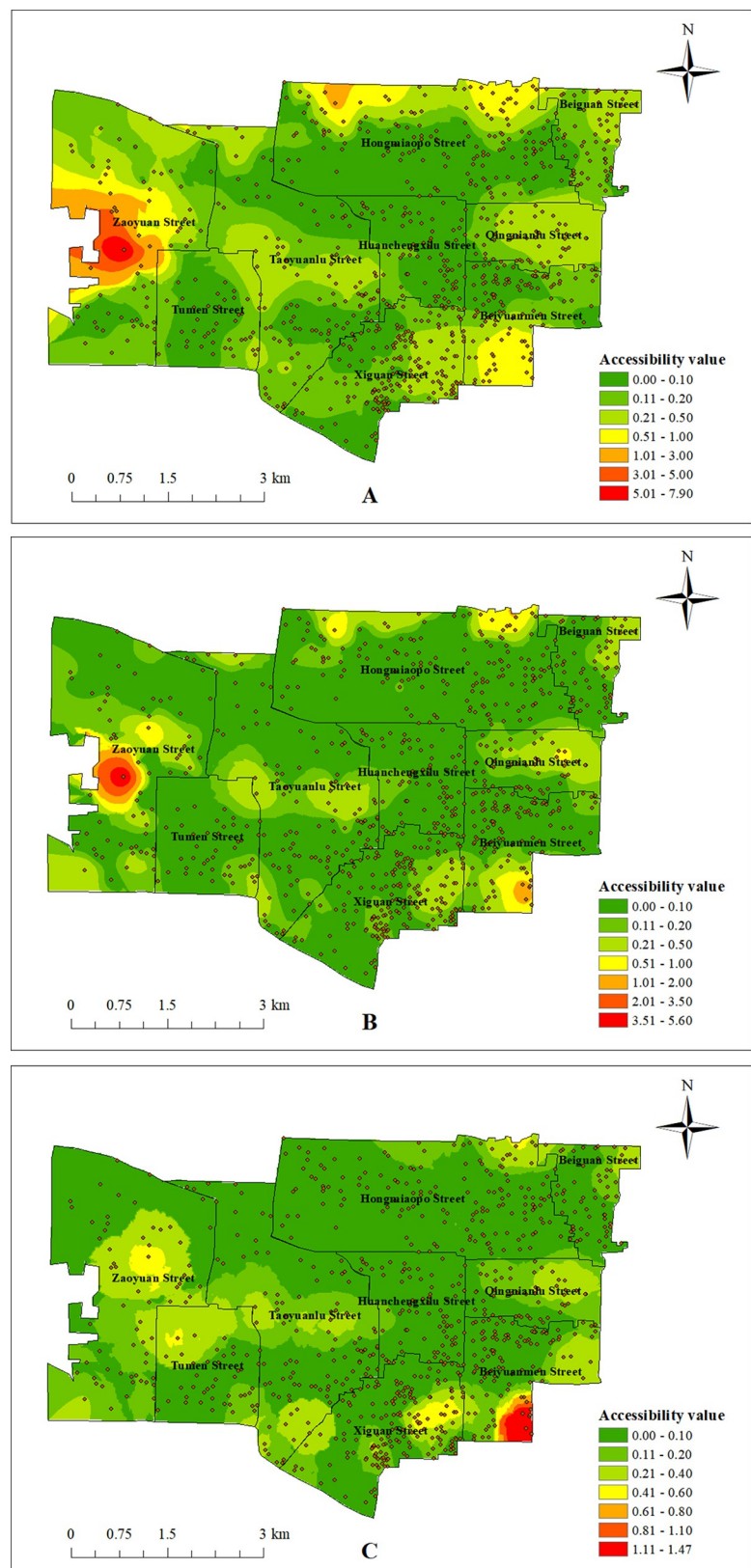

**Fig 9. Accessibility analysis of community home care facilities.** (A) Accessibility value of elderly with normal
mobility walking to community home care facilities; (B) Accessibility value of walking to community home care

facilities for the elderly with mild mobility restriction; (C) Accessibility value of walking to community home care facilities for the elderly with severe mobility restriction. (Self-drawn by the author).

the elderly with severe mobility restriction are generally high, with Gini coefficients greater than 0.4, which are in the 'relatively large disparity' and 'great disparity' levels. The calculation results show that the degree of resource allocation of community home care facilities in Lianhu District is good from the perspective of the elderly with normal mobility, while the degree of resource allocation of community home care facilities is poor and polarized from the perspective of the elderly with mobility restriction. Due to the distribution of facilities, the distribution of the elderly population and the density of the road network, the overall distribution of community home care facilities in Zaoyuan Street, Hongmiaopo Street and Beiyuanmen Street is poor, which is consistent with the results of the accessibility analysis. This indicates that the planning of community home care facilities in Lianhu District does not take into account the disadvantaged group of elderly with mobility restriction, and it reflects to a certain extent that the speed of building community home care facilities in Lianhu District cannot catch up with the speed of population aging.

The Lorenz curves were plotted to visualize the distribution of community home care facility resources to the elderly in each street in Lianhu District, as shown in Fig 10. The curves for the elderly with normal mobility in each street were closest to the absolute equity curve, indicating that the community home care facility resources in each street were more equitably distributed to the elderly with normal mobility. On the other hand, the elderly with mobility restriction do not have fair access to the facility resources, 60% of the elderly with mild mobility restriction in each street get less than 30% of the community home care facility resources, 60% of the elderly with severe mobility restriction get less than 20% of the community home care facility resources, and 50% of the elderly with severe mobility restriction in individual streets do not have access to the community home care facility resources, indicating that the distribution of community home care facility resources among the elderly with mobility restriction in Lianhu District is polarized and there is inequity in the distribution of community home care facility resources.

The location entropy is used to analyze the spatial pattern of the matching between the elderly with normal mobility and the elderly with mobility restriction and the resources of community home care facilities. The location entropy is divided into five levels, and the Street-level location entropy classification map is drawn, as shown in Fig 11.

The location entropy of the streets from the perspective of the elderly with normal mobility is generally higher than that of the elderly with mobility restriction, with more than half of the

**Table 4. Gini coefficient of community home care facilities.**

| Administrative Street | Elderly with normal mobility | Elderly with mild mobility restriction | Elderly with severe mobility restriction |
|---|---|---|---|
| Qingnianlu Street | 0.27 | 0.46 | 0.57 |
| Beiyuanmen Street | 0.48 | 0.62 | 0.76 |
| Beiguan Street | 0.29 | 0.49 | 0.59 |
| Hongmiaopo Street | 0.59 | 0.77 | 0.79 |
| Huanchengxilu Street | 0.35 | 0.59 | 0.72 |
| Xiguan Street | 0.33 | 0.51 | 0.68 |
| Tumen Street | 0.38 | 0.61 | 0.72 |
| Taoyuanlu Street | 0.41 | 0.61 | 0.86 |
| Zaoyuan Street | 0.52 | 0.54 | 0.79 |

**Table 5. Gini coefficient level established by the United Nations Development Program.**

| Gini coefficient | Index ranking | Distribution degree |
|---|---|---|
| Less than 0.2 | very low | Height average |
| 0.2-0.29 | low | Comparative average |
| 0.3-0.39 | mid | Relative reasonable |
| 0.4-0.59 | high | Relatively large disparity |
| More than 0.6 | very high | Great disparity |

streets having a location entropy greater than 1.0, and the highest value of location entropy is the location entropy of Zaoyuan Street from the perspective of the elderly with normal mobility, with a value of 2.04, as shown in Table 6. The entropy values of the streets from the perspective of the two types of elderly with mobility restriction are low, with 45.39% of the streets with mild mobility restriction having entropy less than 0.5 and 76.05% of the streets with severe mobility restriction having entropy less than 0.5. The entropy values of all streets from the perspective of these two types of elderly are only 'very low' and 'low', as shown in Table 7 and Table 8. In terms of the overall elderly population, the resources of community home care facilities in the administrative streets of the eastern region of Lianhu District are generally more balanced. However, for the elderly with mobility restriction, there is a relative lack of resources for community aging-in-place facilities in Lianhu District, and there is a spatial mismatch between the elderly population with mobility restriction and the facilities. Among them, the Huanchengxilu Street and Tumen Street have the most scarce resources of community home care facilities from the overall elderly perspective, and are important streets to improve the configuration and layout of community home care facilities in the future.

## 4. Discussion

### 4.1 Contributions to research analysis methods

This paper provides an analytical framework for the equity of access to community home care facilities resources for elderly with different mobility abilities. Based on the seventh national census data, community POI data, and map vector data, the coverage, accessibility, and spatial pattern of community-based home care facilities from the perspective of elderly with different mobility abilities are analyzed using GIS spatial analysis techniques, and the research system is scientific and operable. The research methods selected are the Kernel Density-type 2SFCA, the Gini coefficient method, the Lorenz curve method, and the Locational entropy method, which are commonly used in the evaluation of accessibility and equity of public service facilities. We have innovative points in other aspects. On the one hand, the traditional research on the accessibility of senior care facilities is aimed at the elderly as a whole and does not consider the influence of physiological differences of the elderly on the results of accessibility analysis. In contrast, the actual elderly policy is biased toward the disadvantaged groups among the elderly, and the user groups of the elderly facilities include the semi-disabled and disabled elderly. Therefore, this paper classifies elderly people according to their walking speed into elderly with normal mobility, elderly with mild mobility restriction and elderly with severe mobility restriction, and analyzes the equity in walking access to community home care facilities services from the perspective of these three groups of elderly people. The results of the analysis show that under the current construction of community home care facilities in Lianhu District, there is inequity in access to community home care facilities between the elderly with normal mobility and the elderly with mobility restriction. On the other hand, we consider the principle of facility sharing when calculating the accessibility. Although the walking speed of

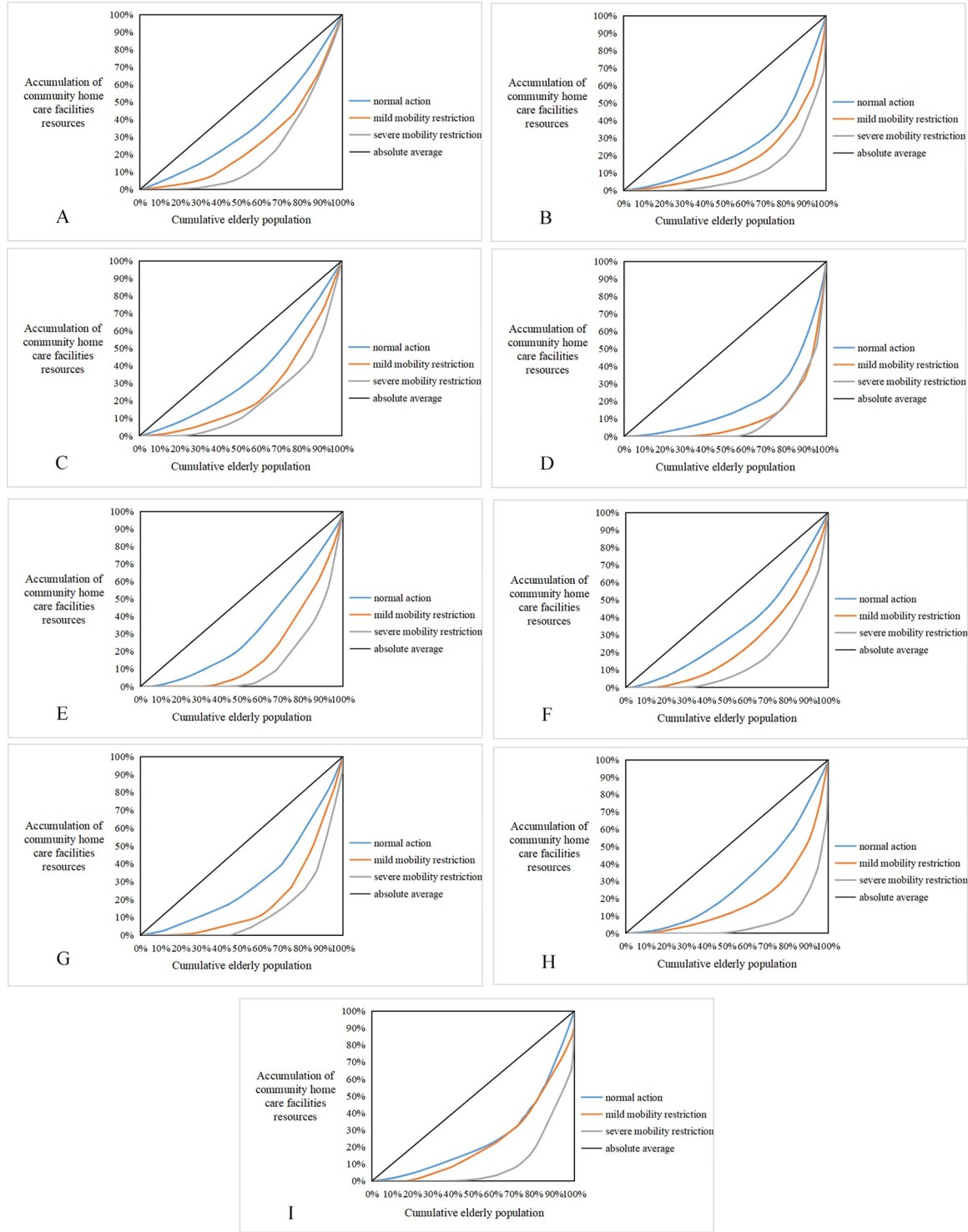

**Fig 10. Lorenz curve of resource allocation of community home care services.** (A) Qingnianlu Street; (B) Beiyuanmen Street; (C) Beiguan Street; (D) Hongmiaopo Street; (E) Huanchengxilu Street; (F) Xiguan Street; (G) Tumen Street; (H) Taoyuanlu Street; (I) Zaoyuan Street. (Self-drawn by the author).

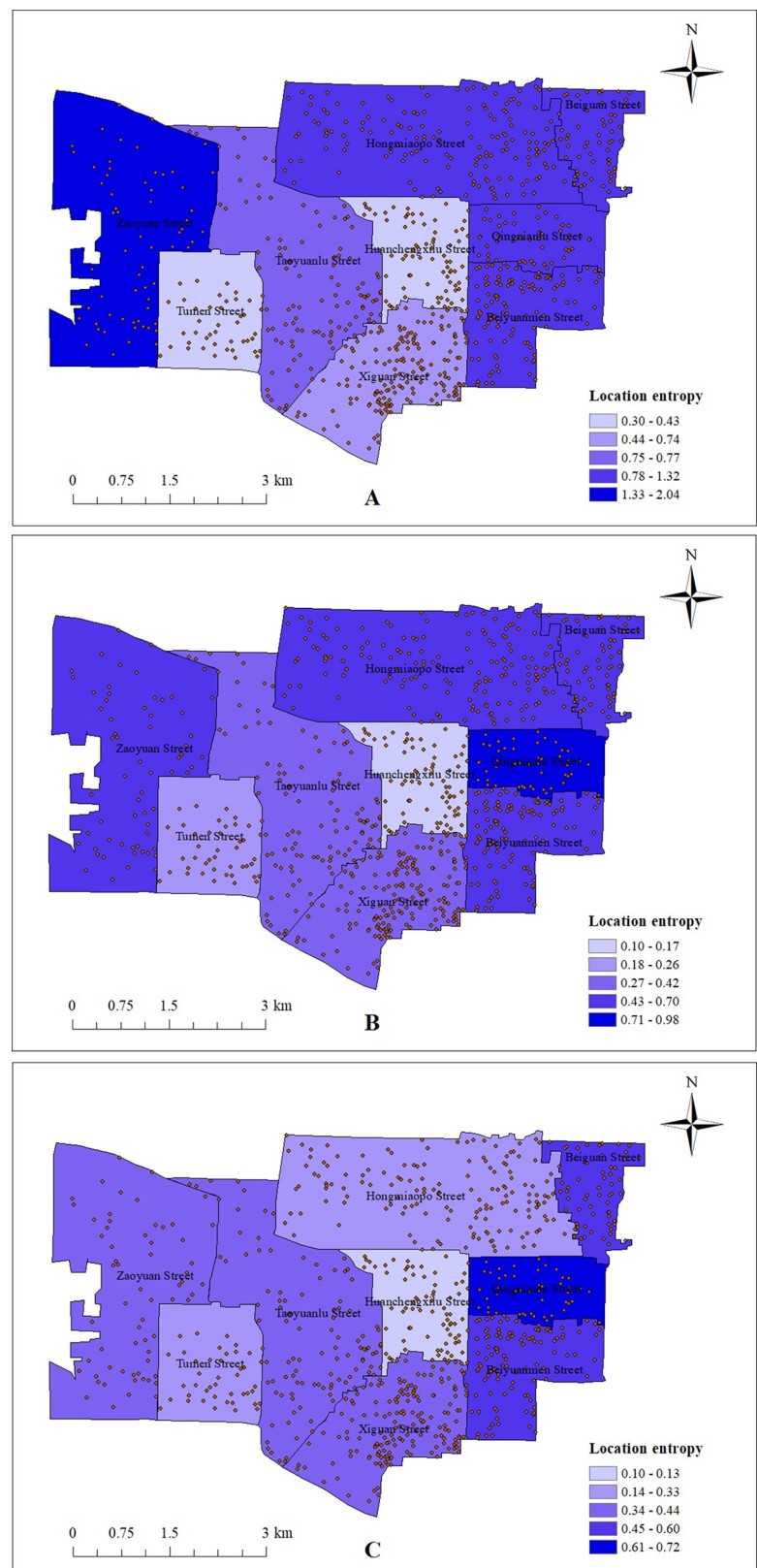

**Fig 11. Location entropy of community home care facilities.** (A) Location entropy from the perspective of elderly with normal mobility; (B) Location entropy from the perspective of the elderly with mild mobility restriction; (C) Location entropy from the perspective of the elderly with severe mobility restriction. (Self-drawn by the author).

**Table 6. Statistics on the number and proportion of the elderly with normal mobility in the entropy sub-grade of the community home care facility area.**

| Level | Location entropy | Number of elderly with normal mobility | proportion |
|---|---|---|---|
| Extremely low | 0.0-0.5 | 24644 | 16.90% |
| Low | 0.5-1.0 | 41562 | 28.50% |
| medium | 1.0-1.5 | 59967 | 41.11% |
| High | 1.5-2.0 | 0 | 0 |
| Extremely high | 2.0-2.5 | 19677 | 13.49% |

the three types of elderly people is different, they are in the same space to access the facility resources, so we calculate the accessibility of the three types of elderly people to access the community home care facilities resources at the same time under their different travel intentions, thus making the results of the equity analysis more accurate. The different extreme travel distances of the elderly reflect the variability of their access to community home care facilities resources, and also show that the reason for this variability is the walking speed of these three types of elderly people, and to reduce this variability, it is necessary to increase the density of community home care facilities, adjust the layout of facilities, and strengthen the road network construction. This study has some reference significance for the study of the equity of elderly care facilities.

## 4.2 Contribution to the optimization of the configuration and layout of community home care facilities

In the context of the continuous deepening of aging, the configuration and layout of urban senior care facilities are facing an important test. Some elderly people in the '15-minute elderly care circle' have access to community elderly care facilities that are much higher than the average water in the area, while some elderly people have little or no access to community elderly care facilities, which is caused by the dense elderly population and the unreasonable layout of facility configuration. This study can provide the following help for the optimization of the layout of community aged home facilities in urban communities. On the one hand, it can identify the areas with weak service capacity of community home care facilities and target these areas for the construction or renovation of community home care facilities to achieve the purpose of spatial matching of facility resources with the aged care needs of the elderly. On the other hand, it is possible to derive the differences in the access of elderly people with different mobility to community home care facilities resources, and adjust the layout of existing community home care facilities according to such differences, so that different elderly groups can have more equitable access to community home care facilities resources. Therefore, a foundation is laid for the research on the configuration and layout of community home care facilities based on elderly with different mobility.

**Table 7. Statistics on the number and proportion of the elderly with mild mobility restriction in the entropy sub-grade of the community home care facility area.**

| Level | Location entropy | Number of elderly with mild mobility restriction | proportion |
|---|---|---|---|
| Extremely low | 0.0-0.5 | 12414 | 45.39% |
| Low | 0.5-1.0 | 14933 | 54.61% |
| medium | 1.0-1.5 | 0 | 0 |
| High | 1.5-2.0 | 0 | 0 |
| Extremely high | 2.0-2.5 | 0 | 0 |

**Table 8. Statistics on the number and proportion of the elderly with severe mobility restriction in the entropy sub-grade of the community home care facility area.**

| Level | Location entropy | Number of elderly with severe mobility restriction | proportion |
|---|---|---|---|
| Extremely low | 0.0-0.5 | 6933 | 76.05% |
| Low | 0.5-1.0 | 2183 | 23.95% |
| medium | 1.0-1.5 | 0 | 0 |
| High | 1.5-2.0 | 0 | 0 |
| Extremely high | 2.0-2.5 | 0 | 0 |

### 4.3 Limitations and future research

There are some limitations and shortcomings in this study. First, due to the limitation of data acquisition, we found and marked down the vertical points of each community and community home care facilities with the nearest road network in ArcGIS software, assuming that these points are the import and export of each community and community home care facilities, which is different from the actual situation; second, we did not consider the influence of community home care facilities in adjacent administrative districts on the access of elderly people to facility resources in this study area; third, we only selected Lianhu District as an example, and the analysis results obtained only represent Lianhu District and cannot cover all types of urban administrative units.

In the future, the scope of the study should be expanded. When obtaining the entrance and exit points of each district and community home care facilities, for districts with multiple entrances and exits, the location of community home care facilities around the district should be considered to determine the entrances and exits for seniors on foot, so that the research data can be more accurate. Consideration should also be given to the elderly's access to the resources of community home care facilities in neighboring administrative districts. The residence of the elderly located at the borders of administrative districts may be closer to the senior care facilities in neighboring administrative districts, and according to the elderly's willingness to travel, they are more willing to go to the community home care facilities in neighboring administrative districts, and this behavior will reduce the pressure of senior care service supply in this administrative district, but at the same time will increase the pressure of senior care service supply in another administrative district, which will have an impact on the urban This behavior will reduce the pressure on the supply of senior care services in this borough, but at the same time will increase the pressure on the supply of senior care services in another borough, which will have an impact on the configuration and layout of urban senior care facilities. Therefore, the results of this study can be compared with the results of the study after the inclusion of community home care facilities in neighboring boroughs, and the differences can provide reference for the allocation and layout of community home care facilities among urban boroughs.

### 5. Conclusion

This paper constructs an evaluation framework for the equity of the layout of community home care facilities. Firstly, by using GIS spatial analysis technology, a buffer is set to calculate the coverage rate of four types of community home care facilities from the perspective of the elderly with different mobility, and the quantitative index of the service level of community home care facilities is obtained. Secondly, based on the actual road network, the accessibility of community home care facilities is measured by the Kernel Density-type 2SFCA. The accessibility value is represented by the per capita usable facility area, and the accessibility value is visualized by the Kriging interpolation method. Finally, the Gini coefficient, Lorenz curve and

location entropy method are used to analyze the equity performance of community home care facilities. Overall, this paper draws the following two conclusions:

The spatial matching of elderly with mobility restriction and community home care facilities varies greatly. Equity in the layout of facilities is mainly concerned with the rational degree of the layout of facilities, so that the configuration and layout of facilities are more inclined to the disadvantaged groups. Therefore, many unused houses in old urban areas can be transformed into community home care facilities, which not only reduces the waste of community resources, but also increases the density of facilities, so that elderly with mobility restriction can more easily access the elderly resources in their living environment. At the same time, China's community home care funding has been based on state funding and community fundraising, with social development, the needs of the elderly services become diversified, community home care facilities hardware is in urgent need of upgrading, relying only on government investment has been unable to meet the development of community home care service facilities, the cost of new elderly facilities is too large, the transformation of public space is greatly reduced capital expenditure, can maximize the community home care service funds to the elderly. As Bai *et al*. stated, tapping into existing community unused resources and encouraging the use of social units and unused houses of residents' families on the first floor of buildings to establish community-based elderly care facilities reduces financial and land pressure [31]. The government document also provides standards for community home care facilities, stating that new communities should build community home care facilities at a rate of 15 to 20 square meters per 100 households, and established communities should build community home care facilities through purchase, replacement and lease.

Road network density is one of the important factors influencing the accessibility of community home care facilities. In the 15-minute elderly care circle, walking is the main way for elderly people to travel, and the construction of a walking transportation network system is particularly important for elderly people who access community home care facilities on foot. The road network density in some areas of Tumen Street, Huanchengxilu Street, Taoyuanlu Street, and Hongmiaopo Street in Lianhu District is too low, and the distribution of facilities is uneven. The road network of these streets should be improved by increasing the number of road branches and opening up the cut-off roads to increase the accessibility of the road network. As Zhang *et al*. suggest, street connectivity can be improved by connecting dead-end streets or completing sidewalks [32]. At the same time, age-appropriate renovation measures such as road accessibility and anti-slip design are taken to make travel safer and more convenient for the elderly with different mobility, to achieve a fair allocation of community home care facilities and to be more caring for the disadvantaged groups.

## Supporting information

**S1 File. Community home care facilities and elderly population data.**
(ZIP)

**S2 File. Equity analysis data.**
(ZIP)

## Author Contributions

**Conceptualization:** Wenze Ning.

**Data curation:** Yi Yang.

**Formal analysis:** Yi Yang.

**Funding acquisition:** Wenze Ning.

**Methodology:** Yi Yang.

**Project administration:** Wenze Ning.

**Software:** Yi Yang.

**Writing – original draft:** Wenze Ning, Yi Yang, Mei Lu, Xiaokang Han.

**Writing – review & editing:** Yi Yang.

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
