## [Decision Letter · Decision Letter 0]

31 Aug 2022

PONE-D-22-19730Equity in walking access to community home care facility resources for elderly with different mobility: A case study of Lianhu District, Xi'anPLOS ONE

Dear Dr. Lu,

Thank you for submitting your manuscript to PLOS ONE. After careful consideration, we feel that it has merit but does not fully meet PLOS ONE’s publication criteria as it currently stands. Therefore, we invite you to submit a revised version of the manuscript that addresses the points raised during the review process.

Please try to improve your paper and respond to all the reviewers' comments.

We look forward to receiving your revised manuscript.

Kind regards,

Quan Yuan, Ph.D.

Academic Editor

PLOS ONE

Journal Requirements:

2. In order to meet journal requirements for reporting and reproducibility, at this time we request that you please update the Methods section to report the original source of the data and the methods used to collect it in sufficient detail for another researcher to access the same data. Please ensure that you include a statement specifying whether the collection and analysis method complied with the terms and conditions of the data source."

"This research was funded by the the Key Research of  Department of Education of Shaanxi Provincial  (21JZ034), and the Ministry of Education Humanities and Social Sciences Planning Foundation of China (21YJA630092)."

5. We note that Figures 1, 2 3, 4 and 5 in your submission contain map images which may be copyrighted. All PLOS content is published under the Creative Commons Attribution License (CC BY 4.0), which means that the manuscript, images, and Supporting Information files will be freely available online, and any third party is permitted to access, download, copy, distribute, and use these materials in any way, even commercially, with proper attribution. For these reasons, we cannot publish previously copyrighted maps or satellite images created using proprietary data, such as Google software (Google Maps, Street View, and Earth). For more information, see our copyright guidelines: http://journals.plos.org/plosone/s/licenses-and-copyright.

a. You may seek permission from the original copyright holder of Figures 1, 2 3, 4 and 5 to publish the content specifically under the CC BY 4.0 license.  

Reviewers' comments:

Reviewer's Responses to Questions

**Comments to the Author**

1. Is the manuscript technically sound, and do the data support the conclusions?

Reviewer #1: Yes

Reviewer #2: Yes

2. Has the statistical analysis been performed appropriately and rigorously? 

Reviewer #1: Yes

Reviewer #2: Yes

3. Have the authors made all data underlying the findings in their manuscript fully available?

Reviewer #1: Yes

Reviewer #2: No

4. Is the manuscript presented in an intelligible fashion and written in standard English?

Reviewer #1: Yes

Reviewer #2: Yes

5. Review Comments to the Author

Reviewer #1: This study investigated the equity in walking access to community home care facilities for elderly people with different mobility abilities in Lianhu District of Xi’an City. The research approach is simple, but the research topic is interesting and meaningful since few studies focused on the perspective of elderly people with different physiological differences. The reviewer has a few suggestions for the authors that can help improve the paper.

1. On Page 5, there are “121” community home care service stations, which is not consistent with “117” in Table 1.

2. In subsection 2.2.2, it does not clear how to calculate the number of elderly population in the community from the street-level aging rate.

3. In subsection 3.1, the authors should specify the measurement method of gait speed, which is the key to measuring the walking time.

4. In subsection 3.1, the author needs to further explain the sampling method to prove that the sample is representative. In addition, why not use the percentage of elderly with different mobility abilities in these obtained samples?

5. On Page 13, what are the criteria for the coverage rate of community home care facilities to be considered to be low?

6. On Page 14, are walkability and accessibility the same concept? If yes, a unified word is recommended.

7. On Page 20, “The location entropy is used to analyze the spatial pattern of the matching between the elderly with normal mobility and the elderly with and the resources of community home care facilities.” This sentence seems to have missed something after the second “with”.

8. In section 4, the similarities and differences with previous studies should be further discussed. The object of Zhao’s study is lower-aged elderly people, which is not comparable with this paper.

9. Some limitations and future work should be provided.

Reviewer #2: Comments: This paper analyzes the equity in walking access to community home care facilities for elderly people with different mobility abilities in Lianhu District of Xi’an City, taking the resources of community home care facilities as the research object, and there are some problems as shown below.

Question1: In the introduction, the author just lists the existing references, and a detailed summary of the references is needed. At the same time, a brief description of the rest of the paper should be added at the end of the introduction.

Question2: In the materials and methods, the authors just used the existing methods (Kernel Density-type two-step floating catchment area method; Gini coefficient method and Lorenz curve method; Location entropy) to analyze the problem, and do not mention the innovation or improvement of these methods.

Question3: In the results, the statistical results of many data are listed, but each result lacks corresponding discussion and result analysis.

Question4: In the discussion, an analytical framework for the evaluation of the equity of the layout of community home care facilities under the perspective of mobility is provided, however, this part is basically the result of some qualitative analysis, lack of quantitative data discussion.

Question5: In the conclusion, this paper constructs an evaluation framework for the equity of the layout of community home care facilities, however, the conclusion part is too long, some descriptions can be put into other modules in the paper.

6. PLOS authors have the option to publish the peer review history of their article (what does this mean?). If published, this will include your full peer review and any attached files.

Reviewer #1: No

Reviewer #2: No

---

## [Author Response · Author response to Decision Letter 0]

4 Oct 2022

Dear Editors:

We gratefully appreciate the editors and all reviewers for their time spend making positive and constructive comments. These comments are all valuable and helpful for revising and improving our manuscript entitled “Equity in walking access to community home care facility resources for elderly with different mobility: A case study of Lianhu District, Xi’an” (ID: PONE-D-22-19730), as well as the important guiding significance to our researches.

We have studied comments carefully and have made correction which we hope meet with approval. Revised portion are marked in red in the revised manuscript. The summary of corrections and the responses to reviewer’s comments are listed in the Response to Reviewers.

The following three notes have been added as requested by the journal.

For the data sources and collection methods, we add a description in Section 2 of the article and state that all data and methods comply with the terms and conditions of the data source.

The roles played by the funders in the study are described below. This research was funded by the Key Research of Shaanxi Provincial Department of Education (21JZ034), and the Ministry of Education Humanities and Social Sciences Planning Foundation (21YJA630092). The funder of the Ministry of Education Humanities and Social Sciences Planning Foundation (21YJA630092) had no role in study design, data collection and analysis, decision to publish, or preparation of the manuscript. The funder of the Key Research of Shaanxi Provincial Department of Education (21JZ034) supported writing the paper.

The copyright of the images we submit is explained below. The map images we use are sourced from public interest websites and are not copyrighted. We have added a description of the data sources in section 2.2 of the article.

Thank you and best regards.

Yours sincerely,

Mei Lu

Corresponding author:

Name: Mei Lu

E-mail: lumei@xauat.edu.cn

---

## [Decision Letter · Decision Letter 1]

25 Oct 2022

Equity in walking access to community home care facility resources for elderly with different mobility: A case study of Lianhu District, Xi'an

PONE-D-22-19730R1

Dear Dr. Lu,

We’re pleased to inform you that your manuscript has been judged scientifically suitable for publication and will be formally accepted for publication once it meets all outstanding technical requirements.

Kind regards,

Quan Yuan, Ph.D.

Academic Editor

PLOS ONE

Additional Editor Comments (optional):

Reviewers' comments:

Reviewer's Responses to Questions

**Comments to the Author**

1. If the authors have adequately addressed your comments raised in a previous round of review and you feel that this manuscript is now acceptable for publication, you may indicate that here to bypass the “Comments to the Author” section, enter your conflict of interest statement in the “Confidential to Editor” section, and submit your "Accept" recommendation.

Reviewer #1: All comments have been addressed

Reviewer #2: All comments have been addressed

2. Is the manuscript technically sound, and do the data support the conclusions?

Reviewer #1: Yes

Reviewer #2: Partly

3. Has the statistical analysis been performed appropriately and rigorously? 

Reviewer #1: Yes

Reviewer #2: Yes

4. Have the authors made all data underlying the findings in their manuscript fully available?

Reviewer #1: No

Reviewer #2: No

5. Is the manuscript presented in an intelligible fashion and written in standard English?

Reviewer #1: Yes

Reviewer #2: Yes

6. Review Comments to the Author

Reviewer #1: The revised paper has responded my concerns and addressed the proposed comments and suggestion. I recommend this paper for publication.

Reviewer #2: I have no other comments, the authors have answered my questions very well, I hope this manuscript can be accepted.

7. PLOS authors have the option to publish the peer review history of their article (what does this mean?). If published, this will include your full peer review and any attached files.

Reviewer #1: No

Reviewer #2: No

---

## [Editor Report · Acceptance letter]

17 Nov 2022

PONE-D-22-19730R1 

Equity in walking access to community home care facility resources for elderly with different mobility: A case study of Lianhu District, Xi’an 

Dear Dr. Lu:

I'm pleased to inform you that your manuscript has been deemed suitable for publication in PLOS ONE. Congratulations! Your manuscript is now with our production department. 

Kind regards, 

on behalf of

Dr. Quan Yuan 

Academic Editor

PLOS ONE